# Identifying Predictors Associated with Risk of Death or Admission to Intensive Care Unit in Internal Medicine Patients with Sepsis: A Comparison of Statistical Models and Machine Learning Algorithms

**DOI:** 10.3390/antibiotics12050925

**Published:** 2023-05-18

**Authors:** Antonio Mirijello, Andrea Fontana, Antonio Pio Greco, Alberto Tosoni, Angelo D’Agruma, Maria Labonia, Massimiliano Copetti, Pamela Piscitelli, Salvatore De Cosmo

**Affiliations:** 1Department of Medical Sciences, Fondazione IRCCS Casa Sollievo della Sofferenza, 71013 San Giovanni Rotondo, Italy; grecoantonio91@gmail.com (A.P.G.); angelo.dagruma480@gmail.com (A.D.); ap.piscitelli@operapadrepio.it (P.P.); s.decosmo@operapadrepio.it (S.D.C.); 2Unit of Biostatistics, Fondazione IRCCS Casa Sollievo della Sofferenza, 71013 San Giovanni Rotondo, Italy; a.fontana@operapadrepio.it (A.F.); m.copetti@operapadrepio.it (M.C.); 3Department of Internal Medicine and Gastroenterology, Fondazione Policlinico Universitario “A. Gemelli” IRCCS, 00168 Rome, Italy; alberto.tosoni@policlinicogemelli.it; 4Unit of Microbiology, Fondazione IRCCS Casa Sollievo della Sofferenza, 71013 San Giovanni Rotondo, Italy; m.labonia@operapadrepio.it

**Keywords:** sepsis, machine learning, prognostication, internal medicine, SOFA

## Abstract

*Background:* Sepsis is a time-dependent disease: the early recognition of patients at risk for poor outcome is mandatory. *Aim:* To identify prognostic predictors of the risk of death or admission to intensive care units in a consecutive sample of septic patients, comparing different statistical models and machine learning algorithms. *Methods:* Retrospective study including 148 patients discharged from an Italian internal medicine unit with a diagnosis of sepsis/septic shock and microbiological identification. *Results:* Of the total, 37 (25.0%) patients reached the composite outcome. The sequential organ failure assessment (SOFA) score at admission (odds ratio (OR): 1.83; 95% confidence interval (CI): 1.41–2.39; *p* < 0.001), delta SOFA (OR: 1.64; 95% CI: 1.28–2.10; *p* < 0.001), and the alert, verbal, pain, unresponsive (AVPU) status (OR: 5.96; 95% CI: 2.13–16.67; *p* < 0.001) were identified through the multivariable logistic model as independent predictors of the composite outcome. The area under the receiver operating characteristic curve (AUC) was 0.894; 95% CI: 0.840–0.948. In addition, different statistical models and machine learning algorithms identified further predictive variables: delta quick-SOFA, delta-procalcitonin, mortality in emergency department sepsis, mean arterial pressure, and the Glasgow Coma Scale. The cross-validated multivariable logistic model with the least absolute shrinkage and selection operator (LASSO) penalty identified 5 predictors; and recursive partitioning and regression tree (RPART) identified 4 predictors with higher AUC (0.915 and 0.917, respectively); the random forest (RF) approach, including all evaluated variables, obtained the highest AUC (0.978). All models’ results were well calibrated. *Conclusions:* Although structurally different, each model identified similar predictive covariates. The classical multivariable logistic regression model was the most parsimonious and calibrated one, while RPART was the easiest to interpret clinically. Finally, LASSO and RF were the costliest in terms of number of variables identified.

## 1. Introduction

According to Sepsis-3 consensus, sepsis is “a life-threatening organ dysfunction caused by a dysregulated host response to infection, and septic shock is a subset of sepsis in which underlying circulatory and cellular/metabolic abnormalities are profound enough to increase mortality” [1]. Despite a global decline in its incidence and mortality, sepsis still represents one of the main causes of health loss worldwide [2]. Moreover, sepsis contributes—directly or indirectly—to at least half of global in-hospital deaths [3]. The reduction of the sepsis-associated burden of morbidity and mortality [4] represents a priority for global health services [5].

In the last decades, a shift in the management of septic patients from intensive care units (ICUs) to general internal medicine (IM) wards has been observed [4,6]. Consequently, internists have exerted a great deal of effort to understand the pathophysiology of this complex phenomenon and improve the management of such patients in a non-intensive environment [4,7]. As a result, the literature on sepsis cases from non-ICU wards is constantly growing [8,9,10,11,12], and the outcome of septic patients managed in non-ICU settings has significantly improved compared to the past [13].

The main causes at the basis of the dramatic epidemiology of sepsis are related to patients’ aging, increased prevalence of frailty, comorbidities [14], immune dysfunction [15], and increasing multidrug resistance [16]. These cofactors account for the exponential increase of sepsis incidence among elderly patients.

It is well recognized that sepsis represents a heterogeneous clinical syndrome [17] and a time-dependent disease [7]: thus, the early recognition of septic patients and the use of a standardized approach are pivotal [13]. However, given that general wards lack vital-function monitoring, the identification of patients at risk for poor outcome is one of the main goals in order to optimize treatment strategies. In addition, clinical characteristics of IM patients can differ from those of ICU patients; therefore, data from the literature, mostly derived from critical care medicine, may not be generalizable to noncritical settings [4]. At present, clinical scores [8,9], biomarkers (i.e., procalcitonin) [18,19], active surveillance cultures [20], or their combination [21] are used in IM settings to stratify patients according to their risk for clinical deterioration (i.e., admission to ICU or death). However, despite the fact that guidelines suggest using clinical scores and biomarkers to stratify patients according to their severity [1], real-world studies have showed contrasting results in different clinical settings [8,9,10,11]. Moreover, given the heterogeneity of patients admitted to IM wards and the relatively small amount of information derived from noncritical settings, the optimal prognostication strategy to apply to these patients is still to come.

The aim of the present study was to identify prognostic predictors in a consecutive sample of septic patients admitted to an internal medicine unit, and to compare different statistical models and machine learning algorithms to predict the risk of death or admission to an intensive care unit.

## 2. Patients and Methods

### 2.1. Patients

This retrospective study was conducted within the Internal Medicine Unit (for a total of 80 inpatients) of the Research Hospital “Casa Sollievo della Sofferenza”, located in San Giovanni Rotondo, Apulia Region, Southern Italy. A team composed of physicians, internal medicine residents, and trainee clinicians revised the clinical charts of the patients admitted between 1 January 2018 and 31 December 2019. The discharge diagnoses of sepsis and/or septic shock were matched with blood culture results obtained from the microbiology laboratory.

Inclusion criteria were: (i) discharge diagnosis of sepsis and/or septic shock according to the Sepsis-3 definition [1]; (ii) positive bloodstream cultures (microbiological identification). Exclusion criteria were: (i) blood culture positivity for contaminating microorganisms (e.g., coagulase negative staphylococci, propionibacterium, corynebacterium, or micrococcus); (ii) blood culture positivity in the absence of clinical criteria of sepsis/septic shock and/or antibiotic treatment.

Of the 4189 patients discharged from the IM ward in the evaluated period, 194 (4.6%) (86 (3.9%) in 2018 and 108 (5.4%) in 2019) received a discharge coded diagnosis of sepsis without mechanical ventilation, and 4 (2019) (0.1%) received a discharge coded diagnosis of sepsis with mechanical ventilation. Thus, a total of 198 patients received a diagnosis of sepsis/septic shock. A total of 50 patients (25.2%) who presented the clinical features of sepsis, but for whom the microbiological identification (negative blood cultures) was not available, were excluded from the evaluated sample. The total number of patients with positive blood cultures retrieved by the microbiology lab was 252 (92 for 2018 and 160 for 2019). Of these, 104 (41.3%) patients (31 (12.3%) for 2018, 73 (30.0%) for 2019) were excluded for the reasons illustrated in Figure 1. Thus, a total of 148 patients had clinically relevant positive blood cultures. Therefore, 148 patients (60 in 2018 and 88 in 2019) (58.6% of initially evaluated sample) met the inclusion/exclusion criteria and were included in the present study.

This retrospective clinical study was approved by the Ethics Committee of the Research Hospital “Casa Sollievo della Sofferenza” (15 June 2020).

### 2.2. Methods

Demographic data (e.g., gender, age), risk factors for multi-drug resistant organisms (MDROs) and methicillin-resistant staphylococcus aureus (MRSA) infections (i.e., patients coming from nursing homes, hospitalizations in the previous 90 days, antibiotic treatment within the previous 10 days, diabetes mellitus, chronic renal failure, chronic respiratory failure, dialysis treatment, presence of bladder catheter, central venous catheter with or without peripheral insertion, corticosteroid therapy, and parenteral nutrition), microbiological identification and antibiotic susceptibility, and source of infection (UTI, lungs, abscesses, vascular accesses, etc.), were retrieved from the clinical charts [6,8,22,23].

In addition, relevant clinical data were acquired: systolic (PAS), diastolic (PAD) and mean (MAP) arterial pressure, heart rate (HR), oxygen saturation (SpO_2_), respiratory rate (RR), body temperature (°C), and diuresis. Mental status was assessed using the Glasgow Coma Scale (GCS) and alert, verbal, pain, unresponsive (AVPU) status [24]. Furthermore, some laboratory data, such as bilirubin, creatinine, lactic acid, platelet count, procalcitonin (PCT), partial arterial oxygen and carbon dioxide pressure (PaO_2_, PaCO_2_), fraction of oxygen inhaled (FiO_2_), and PaO_2_/FiO_2_ (P/F) and SpO_2_/FiO_2_ (S/F) ratios, were collected. Clinical and laboratory parameters were used to calculate some of the clinical scores used in the daily clinical practice with predictive or prognostic purposes, such as sequential organ failure assessment (SOFA), quick SOFA (qSOFA), mortality in emergency department sepsis (MEDS) and national early warning score 2 (NEWS2), as previously described [8]. The aforementioned data (laboratory parameters and clinical scores) were evaluated at the time of blood culture collection (which were later found to be positive), usually performed when patients developed criteria for suspected sepsis. Moreover, the same data were re-calculated after 72 h to evaluate their kinetics in order to obtain a variation index over time (delta) [18].

The main outcome of the study was the composite event defined as “death and/or transfer to the Intensive Care Unit (ICU)”, the latter being a direct indicator of clinical worsening.

### 2.3. Statistical Analysis

Patients’ characteristics were reported as mean ± standard deviation (SD) or as absolute and relative frequencies (%) for continuous and categorical variables, respectively. For continuous variables, the symmetry of their distributions was assessed using the skewness index, whereas the assumption of normality was assessed by both the Q-Q plot and the Shapiro–Wilk test. For non-normal (skewed) variables, the median, along with interquartile range (IQR, i.e., first-third quartiles) was reported instead of the mean ± SD. In particular, in the presence of right-skewed distributions, statistical analyses were performed on log-transformed values. Comparisons of the patients’ characteristics were performed with respect to the groups defined by the composite outcome status. For categorical variables, the associations with the composite outcome were assessed, using the chi-square or Fisher exact tests, as appropriate. For normally distributed (or symmetric) continuous variables, the associations with the composite outcome were assessed using the two-sample *t*-test. For right-skewed continuous variables, the associations were assessed using the two-sample *t*-test on log-values or, in the presence of other types of skewness, using the Mann–Whitney *U* test. To identify independent clinical predictors of the outcome at issue, the following four statistical methods were performed and compared: (1) a multivariable logistic model with a stepwise variable selection procedure [25]; (2) a cross-validated multivariable logistic model with the least absolute shrinkage and selection operator (LASSO) penalty [26]; (3) a recursive partitioning and regression tree (RPART) algorithm [27]; (4) a random forest algorithm (RF) [28]. Methods 1 and 2 are classical statistical models, while methods 3 and 4 are machine learning algorithms. As for model 1, the stepwise procedure iteratively finds a particular set of covariates so that in the final model, each included covariate result was statistically significant. In models 1 and 2, risks were reported as odds ratio (OR) along with their 95% confidence interval (CI). Methods 3 and 4 are so called “tree-based algorithms”, where RPART is a classical classification tree, whereas the RF consists in an ensemble of them, where each tree is built on a bootstrap sample from the original dataset. In the RF, about one-third of the observations are left out of the bootstrap sample and are not used in the construction of each tree: this is the so-called out of bag (OOB) data. To split each tree node, if there are M candidate input variables in the dataset, only m << M variables are randomly selected (each time, among the M) to be considered for that split. The tree is grown until the minimum node size is reached (user-defined stop criterion). The OOB data are also used to obtain a variable importance (VIMP) estimate: this is defined as the average over all trees of the difference in prediction error before and after the permutation of the values of each variable in the OOB data within each tree. For each estimated method (i.e., model or algorithm), the list of the most associated predictors was drawn up and its “performance” was assessed. The latter was assessed in terms of (1) discriminatory ability (of the method to allocate a higher probability of having the exitus in dead and/or ICU patients than in living patients) and (2) in terms of calibration (i.e., the ability of the method to allocate predicted probabilities of having the exitus that are aligned with the observed frequencies). The discriminatory ability was assessed by the area under the receiver operator characteristic (ROC) curve (AUC) on these probabilities, along with its 95% CI computed using the DeLong method. Moreover, the method’s calibration was assessed by a nonparametric smoothed calibration curve estimated over a sequence of predicted values. To build the calibration curve, 1000 bootstrap resamples were performed to obtain bias-corrected (overfitting-corrected) estimates of predicted vs. observed values. The plot shows smoothed curves for the original calibration and the optimism-corrected calibration. Perfect calibration should lie on or around a 45° line of the plot. The mean absolute calibration error (MACE) was computed from each calibration plot and refers to the mean difference between the predicted values and the corresponding bias-corrected calibrated values (the lower, the better calibration). Due to its relative low percentage, missing values for some predictors (covariates) in the dataset were imputed to allow each method to be performed using all candidate covariates together. However, the parameter estimates and performance of each method were evaluated in the original (non-imputed) dataset by including only the predictors selected from it, with the exception of the RF, which exploited the contribution made by all included predictors. To impute missing values, multivariate imputation by chained equations algorithm was used. Specifically, 10 chains of multiple imputations were created with 50 iterations per chain using a random forest of 10 trees per iteration. A *p*-value < 0.05 was considered for statistical significance. All statistical analyses were carried out using the R Foundation for Statistical Computing software (ver. 4.0, packages: “tableone”, “mice”, “glmnet”, “rpart”, “randomForestSRC”, “pROC”, “rms”).

## 3. Results

### 3.1. Demographic and Anamnestic Characteristics of Patients at Enrollment

The main demographic and clinical characteristics of the 148 patients evaluated at admission, together with risk factors for MDR infection, are summarized in Table 1.

In particular, 77 (52%) patients were male, the mean age was 70.4 ± 16.2 (range: 18–98) years old. A total of 40 (27.0%) patients had diabetes mellitus, 26 (17.6%) were institutionalized, 48 (32.4%) suffered from active neoplasm, 55 (37.2%) had an end-stage disease. A high proportion of patients showed risk factors for infection and a total of 122 (82.4%) patients were at risk for MDRO/MRSA infection. However, the prevalence of MDRO/MRSA was 27.7%. Infections by Gram-negative bacteria accounted for 40.8%, Gram-positive microorganisms accounted for 46.9%, while fungi accounted for 12.2% of the total.

A total of 37 (25.0%) patients reached the composite outcome (death and/or transfer to ICU). In particular, 33 (22.3%) patients died and 4 (2.7%) patients were admitted to ICU. The time elapsed from admission to the composite outcome was short (median: 7 days, IQR: 4–17 days).

Among the patients who achieved the composite outcome, there was a significantly higher percentage residing in nursing homes (*p* = 0.006), affected by end-stage disease (*p* < 0.001) or respiratory failure (*p* = 0.002), treated with haemodialysis (*p* = 0.015), carrier of bladder catheter (*p* = 0.01), or affected by MDRO/MRSA bacteria (*p* = 0.044) than that of patients who did not achieve the outcome. Moreover, a significantly higher percentage of altered mentation (i.e., A.V.P.U. =V/P/U) (*p* < 0.001) and qSOFA ≥ 2 at admission (*p* = 0.005) was found in this group (Table 1).

Concerning clinical and laboratory parameters, patients reaching the composite outcome showed significantly lower GCS scores (*p* < 0.001), higher FiO_2_ (*p* = 0.001), a lower S/F ratio (*p* = 0.001), and a lower P/F ratio (*p* = 0.002) (Table 2). Moreover, significantly higher creatinine levels (*p* = 0.044) and lower platelet counts (*p* = 0.002) were found at admission in this group. An interesting finding, which was, however, in line with previous observations [8,18], was that mean PCT levels did not significantly differ among the two groups.

Regarding clinical scores, with respect to patients who did not achieve the outcome, those reaching the composite outcome had significantly higher mean scores at SOFA (median: 6 IQR: 5–7 vs. median: 3 IQR: 2–5; *p* < 0.001), qSOFA (median: 1 IQR: 1–2 vs. median: 1 IQR: 0–1; *p* < 0.001), NEWS (median: 9 IQR: 6–10 vs. median: 6 IQR: 4–8; *p* = 0.002), and MEDS (median: 14 IQR: 10–19 vs. median: 8 IQR: 5–12; *p* < 0.001) (Table 2). With regard to the change in values after 72 h (delta), patients reaching the composite outcome showed a lower reduction in terms of delta SOFA (*p* = 0.013), delta qSOFA (*p* = 0.001), and delta-PCT (*p* = 0.002) (Table 2).

### 3.2. Building Classical Statistical Models and Machine Learning Algorithms for Mortality and/or ICU Transfer Risk Prediction

#### 3.2.1. METHOD 1: Multivariable Logistic Model Using the Stepwise Variable Selection Procedure

This model identified three independent predictors of the composite outcome: SOFA score at admission (OR: 1.83; 95% CI: 1.41–2.39; *p* < 0.001), delta SOFA (OR: 1.64; 95% CI: 1.28–2.10; *p* < 0.001), and AVPU (OR: 5.96; 95% CI: 2.13–16.67; *p* < 0.001). This model not only included 3 covariates (with no missing values) but also achieved the highest calibration accuracy (i.e., predicted risk probabilities > 40% quite overlapped the observed ones, reaching the lowest MACE of 0.031), along with a high discriminatory ability (AUC: 0.894; 95% CI 0.840–0.948) (Figure 2, panels: A,B).

#### 3.2.2. METHOD 2: Cross-Validated Multivariable Logistic Model with LASSO Penalty

Using the dataset with imputed missing values, this model identified the following eight independent predictors: SOFA score at admission, delta SOFA, delta qSOFA, MEDS score at admission, delta-PCT, presence of COPD, dialysis, and AVPU. This is due to the fact that the 10-fold cross-validated error (binomial deviance) reached the minimum value, within 1 standard error, at the log-penalty parameter of −3.08 (Appendix A). Using the original dataset (with missing values), these covariates were then included in a classical multivariable logistic model to obtain the parameter estimates and their 95% CI. However, in doing so, serious problems occurred during the estimation of these parameters. It was therefore decided to exclude COPD patients, those on dialysis, and the MEDS score from the set of covariates for the following distinct reasons: (1) no dialysis patients were observed among those who did not experience the composite outcome; (2) only one COPD patient was found among those who experienced the composite outcome; (3) when considered together with the other covariates, the MEDS score lost its association completely, being highly nonsignificant. Following this, the classical logistic model, including all remaining predictors, provided the following estimates: SOFA score at admission (OR: 1.81; 95% CI: 1.30–2.54; *p* < 0.001), delta SOFA (OR: 1.50; 95% CI: 1.06–2.11; *p* = 0.022), delta qSOFA (OR: 1.82; 95% CI: 0.80–4.14; *p* = 0.150); delta-PCT (OR: 1.63; 95% CI: 0.90–2.96; *p* = 0.107), and AVPU (OR: 5.48; 95% CI: 1.35–22.24; *p* = 0.017). Of the 148 patients, 42 were not considered because of missing information on 1 or more covariates. Therefore, the classical logistic model was estimated on 106 subjects, of whom 23 reached the outcome. The LASSO method results were well calibrated in accurately identifying the lowest and highest risk probabilities (i.e., the predicted risk probabilities < 20% and > 80% overlapped fairly closely with those observed, achieving an overall low MACE of 0.044) while still achieving a higher discriminatory ability (AUC: 0.915; 95% CI: 0.848–0.982) than that achieved by the stepwise logistic model (Figure 2, panels: C,D).

#### 3.2.3. METHOD 3: Recursive Partitioning and Regression Tree

To build a pruned classification tree, the setting of a cost-complexity parameter (CP) value is needed. More specifically, CP denotes the factor due to which no attempt is made to split nodes of the tree that do not reduce the overall lack of fit by at least CP. As shown in Appendix A, the “optimal” CP value that produced the minimum cross-validated error was 0.085, returning a pruned classification tree with 5 terminal nodes (Figure 3, panel A). The terminal node with the lowest percentage of patients experiencing the composite outcome (i.e., the “event”) was defined by all patients having a SOFA score at admission < 4 (Node 1: lowest risk). Indeed, in Node 1, there were 70 patients and only 1 of them experienced the event (1.4%). In contrast, the terminal node with the highest percentage of patients experiencing the composite outcome was defined by all patients having a SOFA score at admission ≥ 4 and a MEDS score ≥ 19 (Node 5: highest risk). Indeed, in Node 5, there were 10 patients and all of them experienced the event (100%). Patients with a SOFA score at admission ≥ 4 and a MEDS score < 19 defined three classes of “intermediate risk”. Specifically, patients with delta SOFA < 0 were defined as belonging to Node 4 (i.e., the “lowest intermediate risk” class), whereas those with delta SOFA ≥ 0 and MAP < 71 were defined as belonging to Node 3 (i.e., the “upper intermediate risk” class), and those with delta SOFA ≥ 0 and MAP ≥ 71 were defined as belonging to Node 2 (i.e., the “highest intermediate risk” class). The number of patients (along with % of events) within terminal Nodes 2, 3, and 4 were: 36 (19.4%), 9 (22.2%), and 23 (73.9%), respectively. No missing values were found in the four covariates used to build the tree. The downside of the clinical utility of classifying patients into risk classes, however, is a lower calibration accuracy. Indeed, observed risk probabilities below 60% were overestimated by the algorithm, while those above 60% were underestimated (i.e., reaching the highest MACE of 0.056) while retaining a high discriminatory ability (AUC: 0.917; 95% CI: 0.868–0.967) (Figure 2, panels: E,F).

#### 3.2.4. METHOD 4: Random Forest

The RF algorithm was also performed using the dataset with previously imputed missing values. The VIMP estimates produced by the RF are shown in Figure 3, panel B. For ease of interpretation, VIMP values were scaled from 0 to 100% of the maximum value achieved (relative VIMP) and only predictors with relative VIMP > 10% were shown. It was found that, consistent with the RPART tree, SOFA score at admission provided the highest contribution in discriminating patients with the presence of the composite outcome from those without (relative VIMP = 100%), followed by the MEDS score at admission (78.3%) and GCS (50.1%), while delta qSOFA provided the least contribution to this discrimination (10.4%). The RF slightly overestimated risk probabilities (i.e., the predicted risk probabilities > 20% were higher than the actual probabilities, achieving an overall low MACE of 0.041) but achieved the highest discriminatory ability (AUC: 0.978; 95% CI: 0.957–0.998) (Figure 2, panels: G,H). This is probably due to the fact that the RF actually included all candidate predictors.

## 4. Discussion

The present study shows that SOFA, AVPU, and delta SOFA are independent predictors of poor outcome (e.g., admission to ICU and death), in a cohort of internal medicine patients affected by sepsis. In addition, different statistical models and machine learning algorithms were used to predict the patients’ outcome with high sensitivity, specificity, and calibration accuracy. These models include adjunctive predictive variables such as delta qSOFA, delta PCT, MEDS, MAP and GCS. As shown in Figure 2, the first method, based on multivariable logistic regression analysis, resulted to be the most parsimonious one achieving a good discriminatory accuracy (AUC = 0.894). Method 2 (LASSO) including 5 predictors and method 3 (RPART) including 4 predictors obtained higher AUC values (0.915 and 0.917, respectively), while the highest AUC (0.978) was obtained with the RF (method 4), a machine learning approach including all evaluated variables.

Sepsis represents a complex syndrome with heterogeneous manifestations in different clinical contexts [13]. In this regard, as no single parameter or indicator—considered alone—is capable of making an accurate diagnosis of sepsis [4], several studies have attempted to compute a score that would allow a rapid and accurate diagnosis of sepsis in different clinical settings [9,29]. Similarly, several attempts have been made to identify a set of parameters to stratify patients at higher risk of rapid deterioration or death, leading to often discordant results [21,30]. In a study performed by our research group, it was shown that MEDS scores and vitamin D status independently predicted mortality in a cohort of 88 septic patients admitted to internal medicine wards of a tertiary care urban University Hospital located in Rome, Italy [8]. Moreover, in that cohort of patients, the kinetic of procalcitonin in the first 48–96 h (i.e., delta-PCT) was also able to predict mortality, whereas PCT at baseline was not [18]. A retrospective study by Papadimitriou-Olivgeris and colleagues, conducted on 404 patients with bloodstream infection admitted to an internal medicine setting in Switzerland, identified qSOFA as an independent predictor of mortality, with high negative predicting value. Moreover, qSOFA showed a higher performance than SIRS [9]. This finding was in line with the recommendation of the Sepsis-3 consensus, suggesting the use of qSOFA for the identification of septic patients at risk for clinical deterioration outside the ICU [29]. Moreover, a retrospective study conducted in a Spanish Internal Medicine Unit showed that the implementation of a Sepsis Code Program, compared to the usual clinical practice, was able to improve the patients’ outcome [12]. A retrospective derivation-validation cohort study by Saeed and colleagues showed that mid-regional proadrenomedullin (MR-proADM) at ED admission outperformed routinely used biomarkers and clinical scores in the early identification of ED patients with suspected infection at risk for deterioration. However, this study was conducted in the ED setting and did not evaluate IM patients [31]. Spoto and coworkers have reported that the combination of PCT and MR-proADM could be useful for the etiological identification of septic patients and their risk stratification. However, the optimal cut-off values are still to be identified [32]. The existence of different phenotypes of sepsis, as proposed by Seymour and colleagues through AI data analysis [17], could explain the heterogeneity of these results.

Clinical characteristics of patients evaluated in the present study are in line with the literature, in terms of patients’ age and comorbidities. Despite a high proportion (82.4%) of patients showing risk factors for MDRO/MRSA infection, the observed prevalence of MDRO (27.7%) was in line with recent national data [33]. Moreover, in our sample, a quarter of patients reached the composite outcome, which is in line with what was found in a previous study carried out by our research group on a different cohort of patients (26.1%) [8], but significantly lower than the percentage reported in SEMINA (36.5%), a recently published prospective regional study on the prevalence of sepsis in internal medicine in the Apulia region [11]. Interestingly, the proportion of deaths observed in our sample is similar to that observed in a Spanish cohort of patients, in the subgroup of patients subjected to the implementation of a sepsis code program [12]. In contrast to our previous findings [8], a higher prevalence of Gram-positive than Gram-negative microorganisms was observed. This result could be explained, at least in part, by the high percentage of patients carrying a central venous catheter (Table 1).

A significantly higher proportion of patients reaching the composite outcome was residing in nursing homes, affected by end-stage disease or respiratory failure, treated with haemodialysis, carrier of bladder catheter, or affected by MDRO than those who did not achieve the outcome. These characteristics represent risk factors for sepsis and common features among high-risk septic patients [8,9,10,11,12,20,21]. Moreover, a significantly higher percentage of altered mentation and qSOFA ≥ 2, as well as significantly lower GCS scores, higher FiO_2_, lower S/F and P/F ratios, higher creatinine levels, and lower platelet counts were found among patients showing the poorest outcome. Even these observations confirm the data of the existing literature [8,9,11,20,21,33]. As already shown, the mean PCT levels did not significantly differ from those who did and did not achieve the composite outcome [8,9,18]. As to clinical scores, patients achieving the composite outcome showed higher scores than the other group of patients. Finally, those patients showing the worst outcome showed a poorer variation of SOFA, qSOFA, and PCT over time. Even this last observation is in line with our previous published data and with the current literature [8,9,18].

Clinical and laboratory characteristics served as the basis for a comparative statistical analysis to predict the risk of death or admission to the ICU. Among the four proposed methods, different strengths and weaknesses can be found as well as different practical utilities. The first method, based on a multivariable logistic model, with a high AUC of 89.4% and the highest calibration accuracy, including only 3 independent predictors (i.e., SOFA score at admission, delta SOFA, and AVPU), was the most parsimonious and efficient model. The second method, based on a cross-validated multivariable logistic model with the LASSO penalty, identified 8 independent predictors, of which only 5 were found to be usable (i.e., SOFA score at admission, delta SOFA, delta qSOFA, MEDS, and delta-PCT), reaching a higher AUC of 91.5% and a good calibration accuracy. The most important strength of this statistical model is the fact that the selection of predictors was robust, as it was carried out using an internally validated penalty parameter. The third method, based on RPART, identified four independent predictors of the composite outcome (i.e., SOFA, MAP, delta SOFA, and MEDS). The most important strength of this algorithm is that it is readily usable by the clinician, directly at the patient’s bedside, as it does not require any application or calculator. Despite its simplicity, this model showed one of the highest AUCs (91.7%), although the calibration accuracy was lower. Finally, the random forest algorithm, based on an AI algorithm, included all independent predictors of outcome and clearly achieved the highest discriminatory ability (AUC = 97.8%). The most important strength of this algorithm is that it allows for a robust and internally validated list of the “most important” predictors of the composite outcome. On the other hand, with this algorithm, it is not possible to derive a final model (or formula) based on the most important predictors, as it is the result of a set of independent classification trees.

The machine learning approach is a relatively new statistical technique, and data from the literature based on these methods are still few, particularly in the field of internal medicine. However, the AI approach seems to be promising for prognostication. In addition, the machine learning approach is also used in many s-Health applications, not only for the diagnosis of bacterial sepsis, but also for the diagnosis of glaucoma, Alzheimer’s disease, ICU readmissions, and cataract detection [34]. A study by Zhang and colleagues on more than 3000 ICU patients with sepsis showed that a scoring system derived by using the LASSO technique (referred to as the LASSO score) achieved the best discrimination ability (AUC = 0.772) when compared to routinely used risk scores [35]. Similarly, Taylor and coworkers demonstrated in a sample of ED septic patients that the RF approach (AUC = 0.86) was superior to classical clinical decision rules to predict in-hospital mortality [36]. Moreover, according to Cheng and collaborators, the evaluation of vital signs by convolutional neural networks (CNNs) (AUC = 0.84) was superior to other machine learning techniques (i.e., long short-term memory (LSTM) and RF) in the prediction of mortality of ED septic patients within 6 to 48 h from admission [37]. As in the present study, all of the four models showed a very high discriminatory accuracy. Moreover, excluding the machine learning method, which requires a computer-based approach, all the remaining models are based on clinical scores or laboratory exams that are routinely assessed during the evaluation of septic patients, independently from the clinical setting, and are, therefore, all easy to use. In this regard, it should be underlined that the use of the SOFA score is recommended by the Sepsis-3 task force within ICU [1,29], and it has been validated outside the ICU, confirming its predictivity of the poorest outcome [38]. The MEDS score, originally validated in EDs [39], has been evaluated by our group in an IM setting, showing a good predictivity.

The main limitation of this study is represented by its retrospective design. Moreover, the present results should be confirmed by prospective studies with a larger sample of patients. However, the internal validation of the LASSO and RF algorithms strengthens our observations.

## 5. Conclusions

The management of septic patients represents a challenge for internists. In the noncritical setting, the possibility to identify those patients at the highest risk for the poorest outcome helps physician to intensify the therapeutic approach, or, when not possible, to inform patients and/or relatives about the prognosis [40].

The present study compares, for the first time, different statistical approaches, each one with its own peculiarities, to build prediction tools that are user friendly for clinicians and based on routinely assessed clinical scores, biomarkers, and vital parameters. Although structurally different, each model identifies the same set of predictive covariates. The classical multivariable logistic regression model was the most parsimonious and calibrated one, while RPART was the easiest to interpret clinically. Finally, the LASSO and RF models were the costliest in terms of number of variables identified.

Even if achieving a perfect score remains utopian, the efforts to develop tools to support clinical decisions should be encouraged, especially in the field of sepsis.

## Figures and Tables

**Figure 1 antibiotics-12-00925-f001:**
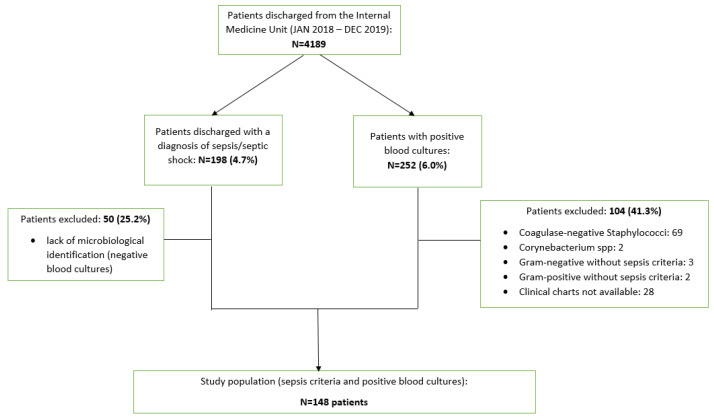
Patients’ flow diagram.

**Figure 2 antibiotics-12-00925-f002:**
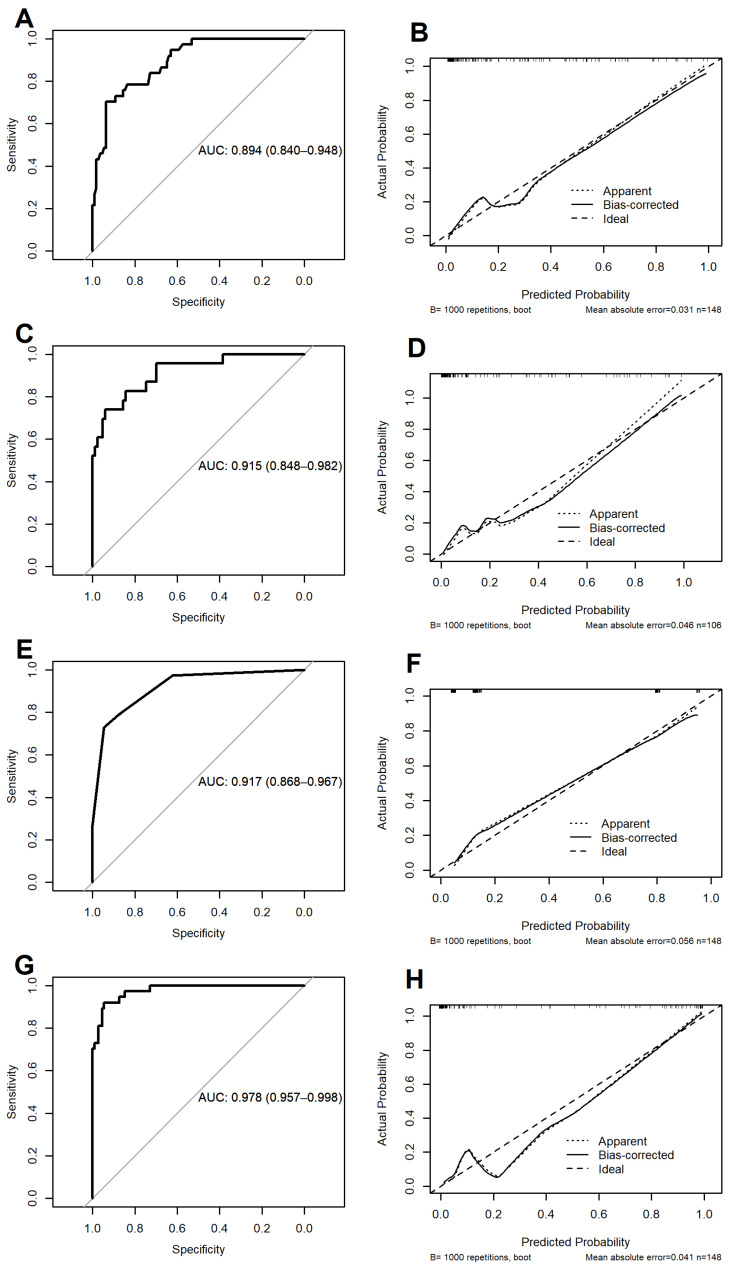
Receiver operator characteristic (ROC) curves and nonparametric smoothed calibration curves of the multivariable stepwise logistic model (**A**,**B**); cross-validated multivariable LASSO logistic model (**C**,**D**); recursive partitioning and regression tree (**E**,**F**), and random forest (**G**,**H**), respectively.

**Figure 3 antibiotics-12-00925-f003:**
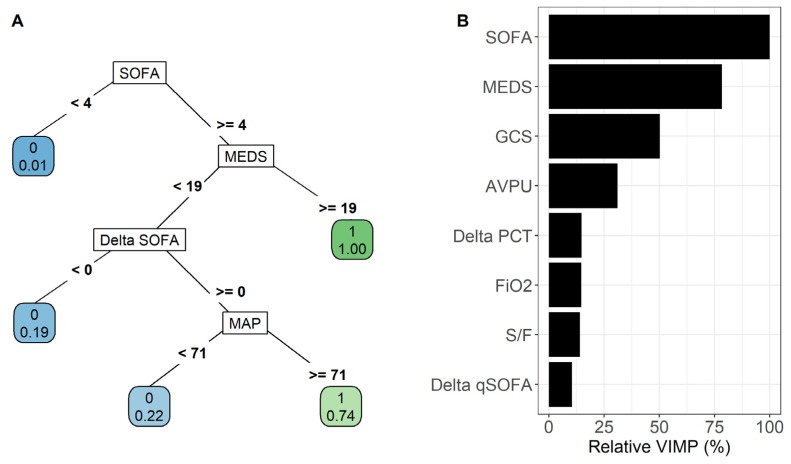
Results from the recursive partitioning and regression tree (RPART) algorithm (panel **A**) and relative variable importance (VIMP) from the random forest algorithm (panel **B**). Abbreviations: SOFA: sequential organ failure assessment; qSOFA: quick SOFA; MEDS: mortality in emergency department sepsis; MAP: mean arterial pressure; GCS: Glasgow coma scale; AVPU: alert, verbal, pain, unresponsive; PCT: procalcitonin; FiO_2_: fraction of oxygen inhaled; S/F: oxygen saturation/FiO_2_. (**A**) The tree defined by RPART identifies patient subgroups at different risk of the composite outcome occurrence (i.e., event). The tree-growing algorithm recursively splits the data into subgroups, choosing the best binary split for each considered variable at issue, to identify the most homogeneous sets within each node and the most heterogeneous ones between the nodes. Splitting variables are shown between branches, while condition sending patients to left or right sibling is on relative branch. Squares indicate subgroups of patients identified by the algorithm (i.e., terminal nodes), whereas circles indicate the splitting nodes. Numbers inside circles represent the predicted event status (0 = No, 1 = Yes) of the entire terminal node (top) and the proportion of patients with the event (bottom) within each terminal node, respectively. If this proportion was greater than 0.5 the predicted event status of the terminal node was “Yes” (otherwise, it was “No). For instance, patients with SOFA score at admission ≥ 4 and MEDS score at admission ≥ 19 defined the class with the highest risk of the event occurrence (i.e., 100% of the patients within this class experienced the event), whereas patients with SOFA score at admission < 4 defined the class with the lowest risk of the event occurrence (i.e., only 1% of the patients within this class experienced the event). (**B**) Predictors identified by the random forest algorithm, sorted from the most to the least “important”, defined by the variable importance measure (VIMP). For ease of interpretation, VIMP values were scaled from 0 to 100% of the maximum value achieved (relative VIMP) and only predictors with relative VIMP > 10% were shown. Consistently with the RPART tree, SOFA score at admission was the most important predictor of the composite outcome, followed by the MEDS score at admission.

**Table 1 antibiotics-12-00925-t001:** Demographic and clinical characteristics of enrolled patients, overall and according to the presence of the composite outcome status (i.e., death and/or admission to intensive care unit).

	Total (N = 148)	Outcome NO (N = 111)	Outcome YES (N = 37)	*p-*Value *
Male gender	77 (52%)	55 (49%)	22 (59.5%)	0.296
Type 2 diabetes mellitus	40 (27.0%)	27 (24.3%)	13 (35.1%)	0.200
Residing in long term facilities	26 (17.6%)	14 (12.6%)	12 (32.4%)	0.006
Age ≥ 65 years	101 (68.2%)	73 (65.8%)	28 (75.7%)	0.262
Active neoplasm	48 (32.4%)	34 (30.6%)	14 (37.8%)	0.417
End-stage illness	55 (37.2%)	32 (28.8%)	23 (62.1%)	<0.001
Hospitalization (previous 90 days)	64 (43.2%)	47 (42.3%)	17 (45.9%)	0.702
COPD	28 (18.9%)	27 (24.3%)	1 (2.7%)	0.003 ^#^
Respiratory failure	42 (28.4%)	24 (21.8%)	18 (48.6%)	0.002
Chronic heart failure	42 (28.4%)	30 (27.0%)	12 (32.4%)	0.528
Chronic kidney failure	38 (25.7%)	25 (22.5%)	13 (35.1%)	0.128
Central venous catether	60 (40.5%)	42 (37.8%)	18 (48.6%)	0.246
Haemodialysis	3 (2.0%)	0 (0%)	3 (8.1%)	0.015 ^#^
Bladder catether	54 (36.5%)	34 (30.6%)	20 (54.1%)	0.010
Parenteral nutrition	45 (30.4%)	30 (27.0%)	15 (40.5%)	0.122
Chronic immunosuppression (including corticosteroids)	45 (30.4%)	24 (21.6%)	8 (21.6%)	1.000
Previous antibiotic treatment (10 days)	55 (37.2%)	39 (35.1%)	16 (43.2%)	0.377
MDRO/MRSA estimated risk	122 (82.4%)	89 (80.2%)	33 (89.2%)	0.212
MDRO/MRSA infection	41 (27.7%)	26 (23.4%)	15 (40.5%)	0.044
AVPU > <A (V/P/U = 1)	47 (31.8%)	22 (19.8)	25 (67.6)	<0.001

Summary statistics are reported as absolute and relative frequencies (%). * *p*-values from chi-square test; ^#^ *p*-values from Fisher exact test. Abbreviations: COPD: chronic obstructive pulmonary disease; MDRO: multidrug-resistant organisms; MRSA: multidrug-resistant staphylococcus aureus; AVPU: alert, verbal, pain, unresponsive.

**Table 2 antibiotics-12-00925-t002:** Laboratory parameters and clinical scores of patients assessed at the time of blood culture collection, overall and according to the presence of the composite outcome status (i.e., death and/or admission toiIntensive care unit).

	Total (N = 148)	Outcome NO (N = 111)	Outcome YES (N = 37)	*p-*Value *
GCS	13.6 ± 2.2	14.2 ± 1.8	12.0 ± 2.4	<0.001
Heart rate (bpm)	96.2 ± 17.0	95.8 ± 16.0	97.4 ± 19.9	0.608
Systolic blood pressure (mmHg)	111.7 ± 22.8	112.8 ± 22.3	108.5 ± 24.5	0.323
Diastolic blood pressure (mmHg)	66.2 ± 12.9	66.8 ± 12.4	64.3 ± 14.5	0.322
Mean blood pressure (mmHg)	81.3 ± 15.3	82.0 ± 14.6	79.0 ± 17.1	0.304
Body temperature (°C)	38.2 ± 1.0	38.3 ± 1.0	38.1 ± 1.1	0.333
FiO_2_	0.2 ± 0.1	0.2 ± 0.1	0.3 ± 0.1	0.001
S/F	408.9 ± 82.8	421.9 ± 74.0	369.8 ± 95.6	0.001
P/F	301.4 ± 85.0	314.1 ± 78.8	263.7 ± 92.4	0.002
PaCO_2_ (mmHg) ^§^	36.5 ± 7.2	37.5 ± 7.4	34.3 ± 6.1	0.098
SpO_2_	93.8 ± 3.6	94.0 ± 3.7	93.2 ± 3.2	0.261
PaO_2_ (mmHg)	70.1 ± 10.7	70.7 ± 10.8	68.4 ± 10.3	0.280
Respiratory rate	19.9 ± 4.7	19.7 ± 4.8	20.7 ± 4.4	0.339
Bilirubin (mg/dL)	0.8 [0.6, 1.5]	0.8 [0.6, 1.4]	0.9 [0.5, 1.8]	0.288 *
Creatinine (mg/dL)	0.9 [0.6, 1.4]	0.8 [0.6, 1.3]	1.2 [0.7, 2.0]	0.044 *
PCT (ng/mL)	4.1 [0.7, 30.6]	4.3 [0.7, 30.6]	3.8 [1.0, 24.9]	0.597 *
PLT (10^3^/µL)	189,500 [120,500, 270,250]	215,000 [137,500, 279,000]	140,000 [98,000, 225,000]	0.002 *
SOFA	4.0 [2.0, 5.0]	3.0 [2.0, 5.0]	6.0 [5.0, 7.0]	<0.001 ^#^
qSOFA	1.0 [0.0, 2.0]	1.0 [0.0, 1.0]	1.0 [1.0, 2.0]	<0.001 ^#^
qSOFA ≥ 2	39 (26.4%)	22 (19.8%)	17 (45.9%)	0.002 °
NEWS2	6.0 [4.0, 9.0]	6.0 [4.0, 8.0]	9.0 [6.0, 10.0]	0.002 ^#^
MEDS	9.0 [6.0, 14.0]	8.0 [5.0, 12.0]	14.0 [10.0, 19.0]	<0.001 ^#^
Delta SOFA	0.0 [−1.0, 1.0]	0.0 [−1.5, 0.0]	0.0 [−1.0, 2.0]	0.013 ^#^
Delta qSOFA	0.0 [−1.0, 0.0]	0.0 [−1.0, 0.0]	0.0 [0.0, 0.0]	0.001 ^#^
Delta MEDS	0.0 [−3.0, 0.0]	0.0 [−3.0, 0.0]	0.0 [−3.0, 1.0]	0.214 ^#^
Delta-PCT %	−1.0 [−1.7, −0.2]	−1.1 [−1.8, −0.4]	−0.3 [−1.3, 0.7]	0.002 *
Delta PLT %	0.0 [−0.2, 0.3]	0.0 [−0.2, 0.3]	−0.1 [−0.5, 0.2]	0.249 *

For normal (continuous) variables, summary statistics are reported as mean ± standard deviation, while for skewed variables they are reported as median along with first and third quartiles. Categorical variables are reported as absolute and relative frequencies (%). * *p*-value from two-sample *t*-test on log transformed values; ^#^ *p*-values from the Mann–Whitney *U* test; ° *p*-value from the chi-square test; ^§^ this variable contains more than 50% of missing values. Abbreviations: FiO_2_: fraction of inspired oxygen; S/F: SpO_2_/FiO_2_ ratio; P/F: Pa = 2/FiO_2_ ratio; PaCO_2_: partial pressure of carbon dioxide; PaO_2_: partial pressure of oxygen; PLT: platelets; PCT: procalcitonin; SOFA: sequential organ failure assessment; qSOFA: quick SOFA; NEWS2: National Early Warning Score; MEDS: mortality in emergency department sepsis.

## Data Availability

Data supporting the reported results may be provided on reasonable request.

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
