# Peer review of "Identifying Predictors Associated with Risk of Death or Admission to Intensive Care Unit in Internal Medicine Patients with Sepsis: A Comparison of Statistical Models and Machine Learning Algorithms"

_antibiotics, 2023, doi:10.3390/antibiotics12050925_

Round 1
Reviewer 1 Report
The manuscript from Mirijello et al is well written and the goal of the study is clear.
The main concern of the study is regarding the use of complex and different statistical methods which results do not add, however, any novelty about methods already used in clinical practice for the diagnosis of sepsis and the patient risk stratification.

Author Response
REPLY to Reviewer #1:
- The manuscript from Mirijello et al is well written and the goal of the study is clear. The main concern of the study is regarding the use of complex and different statistical methods which results do not add, however, any novelty about methods already used in clinical practice for the diagnosis of sepsis and the patient risk stratification.
- REPLY: We thank the Reviewer for the interest in our work and for the time spent in revising our manuscript. Given the continuing evolution of statistical methods and the recent implementation of machine-learning techniques, we performed a comparison between different statistical methods in order to answer the question whether more complex models including a higher number of variables could perform better than classical parsimonious models in terms of risk stratification and prediction of the worst outcome.
- The manuscript “Identifying predictors associated with risk of death or admission to Intensive Care Unit in Internal Medicine patients with sepsis: a comparison of statistical models and machine-learning algorithms” by Mirijello et al. is focused on trying to develop predictive tools to support clinical decision in the field of sepsis using different statistical models. The approach to the topic is quite original since the sepsis is considered the leading cause of death in the world and the clinicians must deal with this issue as quickly as possible. Therefore, the comparison of different statistical models addressed to ameliorate and accelerate the diagnosis of sepsis could be very helpful for clinicians. Although the manuscript is quite clear, I have some concerns to be addressed:
The aim and the rationale for using complex statistical models are not very clear. Why do the authors consider the use of this approach important when more simple and rapid tools are available (i.e. SOFA scores, predictive laboratory biomarkers)? Please, specify.
- REPLY: We thank the Reviewer for this much appreciated comment. The limited information on the utility of clinical scores and biomarkers in IM setting is been highlighted as well as the contrasting results derived from real-word studies (see introduction section, lines 79-83).
- This study does not add any further information on recent approaches to sepsis diagnosis and prognosis (Saeed et al. Crit Care 23:40, 2019; Spoto et al. Microb Pathog 137:103763, 2019). Please, comment this aspect.
- REPLY: according to this Reviewer’s suggestion, we now have reported the suggested papers in the discussion section (lines 420-427, REFERENCES n. 31-32). Although, as stated, our manuscript does not – apparently – add any further information on recent approaches to sepsis diagnosis and prognosis, it is potentially relevant for risk stratification by deepening knowledge from the IM perspective of sepsis.
- The statistical models analysed are not clear enough to readers and the comparison among them does not highlight the main differences and the consequent importance of each one. The authors should explain it better in the discussion.
- REPLY: Thank you for your comment. For each proposed model, a methodological reference has now been included so that interested readers can read it for further details (references 25-28).
- In the conclusions of the study, no information is provided on which of the 4 statistical methods analysed turned out to be the best. They seem only confirm studies already published on the same item. Please, comment this issue.
- REPLY: thank you for your comment: a specific sentence in the conclusion section has now been added (lines 515-521). Each model has its own peculiarities, and it is not possible to determine which of these is the absolute the best.
- In general, the English language should be improved to make the text more readable.
- REPLY: English Language has been revised by Ms. Chiara Di Giorgio, the scientific English proof-reader of our Institute.
Reviewer 2 Report
Authors designed a new approach using Machine Learning to predict the risk of death or admission to Intensive Care Unit admission of patients with Sepsis. The idea and contributions of paper are good. But paper needs some major modifications.
· The abstract and conclusion need to be improved. The abstract must be a concise yet comprehensive reflection of what is in your paper. Please modify the abstract according to “motivation, description, results and conclusion” parts. I suggest extending the conclusions section to focus on the results you get, the method you propose, and their significance. In the method add what analysis was performed.
· I suggest the authors revise Section 1. Please revise the content according to the development of timeline.
· What is the motivation of the proposed method? The details of motivation and innovations are important for potential readers and journals. Please add this detailed description in the last paragraph in section I. Please modify the paragraph according to "For this paper, the main contributions are as follows: (1) ......" to Section I. Please give the details of motivations. In Section 1, I suggest the authors can amend your contributions of manuscript in the last of Section 1.
· The description of manuscript is very important for potential reader and other researchers. I encourage the authors to have their manuscript proof-edited by a native English speaker to enhance the level of paper presentation. There are some occasional grammatical problems within the text. It may need the attention of someone fluent in English language to enhance the readability.
· Please give the details of proposed method for proposed model. I suggest the authors amend the calculation of your size of proposed method and the details are important for proposed method.
· The content of experiments needs to amend related experiments to compare related SOTA in recent three years. I recommend the authors amend related experimental results of proposed method of SOTA according to the published paper in IEEE, Springer, MDPI and Elsevier.
· In the conclusion section, the limitations of this study and suggested improvements of this work should be highlighted.
· Provide a critical review of the previous papers in the area and explain the inadequacies of previous approaches.
· Please cross check the scale of all figures in the manuscript to make them clear and more presentable.
· Include latest references focusing on Machine Learning, Smart Healthcare, and Sepsis.
Moderate revision required
Author Response
REPLY to Reviewer #2:
Authors designed a new approach using Machine Learning to predict the risk of death or admission to Intensive Care Unit admission of patients with Sepsis. The idea and contributions of paper are good. But paper needs some major modifications.
REPLY: We thank the Reviewer for the interest in our work and for the time spent in revising our manuscript.
- The abstract and conclusion need to be improved. The abstract must be a concise yet comprehensive reflection of what is in your paper. Please modify the abstract according to “motivation, description, results and conclusion” parts. I suggest extending the conclusions section to focus on the results you get, the method you propose, and their significance. In the method add what analysis was performed.
- REPLY: We thank the reviewer for this comment. According to the Journal style, we prefer to maintain the present structure (Background, Aim, Methods, Results, Conclusions). However, we have now rewritten the abstract conclusions in order to improve its significance.
- I suggest the authors revise Section 1. Please revise the content according to the development of timeline.
- REPLY: Section 1 was revised. Please see also reply to Reviewer #1, point n.2.
- What is the motivation of the proposed method? The details of motivation and innovations are important for potential readers and journals. Please add this detailed description in the last paragraph in section I. Please modify the paragraph according to "For this paper, the main contributions are as follows: (1) ......" to Section I. Please give the details of motivations. In Section 1, I suggest the authors can amend your contributions of manuscript in the last of Section 1.
- REPLY: The last paragraphs of section 1 have been modified in order to underline the lack of definitive data in the field of sepsis prognostication, in IM setting (lines 79-81).
- The description of manuscript is very important for potential reader and other researchers. I encourage the authors to have their manuscript proof-edited by a native English speaker to enhance the level of paper presentation. There are some occasional grammatical problems within the text. It may need the attention of someone fluent in English language to enhance the readability.
- REPLY: English Language has been revised by Ms. Chiara Di Giorgio the scientific English proof-reader of our Institute.
- Please give the details of proposed method for proposed model. I suggest the authors amend the calculation of your size of proposed method and the details are important for proposed method.
- REPLY: Thank you for your comment. For each proposed model, a methodological reference has now been included so that interested readers can read it for further details.
- The content of experiments needs to amend related experiments to compare related SOTA in recent three years. I recommend the authors amend related experimental results of proposed method of SOTA according to the published paper in IEEE, Springer, MDPI and Elsevier.
- REPLY: we respectfully disagree with the Reviewer, since we prefer not to modify the paper according to this suggestion: the main literature data have been sufficiently discussed.
- In the conclusion section, the limitations of this study and suggested improvements of this work should be highlighted.
- REPLY: we thank the Reviewer for this comment. The last paragraph of the discussion section includes limitations and potential areas of improvement.
- Provide a critical review of the previous papers in the area and explain the inadequacies of previous approaches.
- REPLY: Even in accordance with Reviewer’s 1 point.3 suggestion, we have expanded the discussion section including two more references (REFERENCES n. 31-32) and discussing the limitations of these papers within our field.
- Please cross check the scale of all figures in the manuscript to make them clear and more presentable.
- REPLY: the final check for figures and tables will be performed by the typesetter.
- Include latest references focusing on Machine Learning, Smart Healthcare, and Sepsis.
- REPLY: Thank you for your comment. A reference focusing on machine-learning approaches in smart health has now been included in the Discussion section (ref. n. 35).